# A Two-Step Transcriptome Analysis of the Human Heart Reveals Broad and Disease-Responsive Expression of Ectopic Olfactory Receptors

**DOI:** 10.3390/ijms241813709

**Published:** 2023-09-05

**Authors:** Sadia Ashraf, O. Howard Frazier, Sylvia Carranza, David D. McPherson, Heinrich Taegtmeyer, Romain Harmancey

**Affiliations:** 1Department of Internal Medicine, Division of Cardiology, McGovern Medical School, The University of Texas Health Science Center at Houston, Houston, TX 77030, USA; sadia.ashraf@uth.tmc.edu (S.A.);; 2Texas Heart Institute at Baylor St. Luke’s Medical Center, Houston, TX 77030, USA

**Keywords:** heart failure, G-protein-coupled receptor, olfactory receptor, gene expression

## Abstract

G-protein-coupled receptors (GPCRs) are critical regulators of cardiac physiology and a key therapeutic target for the treatment of heart disease. Ectopic olfactory receptors (ORs) are GPCRs expressed in extra-nasal tissues which have recently emerged as new mediators in the metabolic control of cardiac function. The goals of this study were to profile OR gene expression in the human heart, to identify ORs dysregulated by heart failure caused by ischemic cardiomyopathy, and to provide evidence suggestive of a role for those altered ORs in the pathogenesis of heart failure. Left ventricular tissue from heart failure patients (*n* = 18) and non-failing heart samples (*n* = 4) were subjected to a two-step transcriptome analysis consisting of the quantification of 372 distinct OR transcripts on real-time PCR arrays and simultaneous determination of global cardiac gene expression by RNA sequencing. This strategy led to the identification of >160 ORs expressed in the human heart, including 38 receptors differentially regulated with heart failure. Co-expression analyses predicted the involvement of dysregulated ORs in the alteration of mitochondrial function, extracellular matrix remodeling, and inflammation. We provide this dataset as a resource for investigating roles of ORs in the human heart, with the hope that it will assist in the identification of new therapeutic targets for the treatment of heart failure.

## 1. Introduction

The global burden of heart failure has nearly doubled over the past 3 decades, driven mostly by a surge of risk factors such as hypertension, obesity, and diabetes mellitus [1]. Heart failure currently affects ~6.7 million Americans aged 20 years and older. By 2030, this population is expected to surpass 8 million, and costs for managing the disease will skyrocket to USD 70 billion [2]. Modulation of G-protein-coupled receptor (GPCRs) activity, a pharmacological strategy readily amenable to disease treatment, has yielded some of the most efficient drugs currently available for the management of heart failure [3]. There are over 200 GPCRs expressed in the cardiovascular system [4,5]. Because most of those receptors are often disregarded in genome-wide transcriptome analysis due to low expression levels, we know very little about their potential involvement in disease mechanisms. Therefore, it is very likely that several cardiac GPCRs with relevance to the pathogenesis of heart failure still remain to be identified.

With approximately 400 members in humans (~1000 in mice), olfactory receptors (ORs) constitute the largest family of GPCRs in mammalian organisms [6,7]. Far from being exclusively dedicated to the detection of odorant molecules at the olfactory epithelium, functional ORs have been detected in a multitude of organs and tissues throughout the body, including in the cardiovascular, renal, and metabolic systems [8,9]. Thus, more than 40 human and rodent ORs are expressed in the kidney, with some receptors identified as key regulators of blood pressure, acidemia, and glucose homeostasis [10]. Proof of concept for the presence of functional olfactory receptors in the heart was provided in 2017, when Jovancevic and colleagues reported on the negative chronotropic and inotropic effects mediated by OR51E1 following exposure to circulating medium-chain fatty-acid-activating ligands [11]. While a handful of ORs are reportedly expressed in the normal heart [12], the full extent of cardiac OR gene expression and their potential modulation by heart failure have never been investigated.

The goals of the present study were to generate a high-resolution map of OR gene expression in the human left ventricle, to identify ORs whose expression is altered in response to heart failure caused by ischemic cardiomyopathy, and to provide some evidence suggestive of a potential involvement of those altered ORs in the molecular mechanisms contributing to the pathogenesis of heart failure. To do so, we performed a two-step transcriptomic analysis of left ventricular samples from patients diagnosed with ischemic heart failure (*n* = 18) and non-failing heart samples (*n* = 4). First, the expression of 372 distinct ORs was quantified on custom real-time PCR arrays while simultaneously determining the global cardiac transcriptomic signature of each individual by RNA sequencing (RNA-Seq). Next, gene expression data were merged and the resulting gene expression matrix queried to identify correlations between ORs and other cardiac genes that were found to be differentially regulated between failing and non-failing hearts. Our strategy led to the identification of an unsuspectedly high number of ORs (>160) expressed at significant levels in the human heart, with at least 38 of those receptors differentially regulated with heart failure and potentially involved in the alteration of biological processes such as mitochondrial function, extracellular matrix (ECM) remodeling, and inflammation. We provide this dataset as a resource for investigating roles of ORs in the human heart, with the hope that it will assist in the identification of new therapeutic targets for the treatment of heart failure.

## 2. Results

### 2.1. Altered mRNA Splicing, Proteostasis, and Bioenergetics Are Transcriptomic Features of Heart Failure Caused by Ischemic Cardiomyopathy

Bulk RNA sequencing data were first subjected to a PCA as unsupervised analysis to investigate the differences in global cardiac gene expression in our human population. The PCA score plot of the two first components explained 88.8% of the variation, with the first principal component (PC1) accounting for 49.1% and the second (PC2) for 39.7%. Projection on these two axes clearly separated the four control individuals from the patients diagnosed with ischemic cardiomyopathy (Figure 1A). Hierarchical clustering of pairwise Pearson r correlation matrix for all 22 individuals confirmed greater dissimilarities between control and failing heart samples (Figure 1B). Gene set enrichment analysis (GSEA) of the 4155 genes found to be differentially regulated between failing and control groups (limma method; *p* < 0.05) revealed that RNA metabolism, protein homeostasis, and bioenergetics were the top biological processes dysregulated by heart failure caused by ischemic cardiomyopathy (Figure 1C). These data are confirmatory of the diagnosis for each individual and further demonstrate that control individuals and heart failure patients can be separated based on their global left ventricular transcriptomic signature.

### 2.2. Cardiac Olfactory Receptor Signaling Is Altered with Heart Failure

The GSEA also identified the expression and translocation of ORs as a biological process significantly altered between control and failing hearts (*p* = 0.01). Out of the 21 genes differentially regulated in the olfactory signaling pathway, 9 encoded proteins critical for the transcription (LDB1, LHX2, and EBF1), intracellular transport (REEP1), and signaling (ADCY3, CNGA4, GNB1, GNAL, and ANO2) of ORs. The other 12 genes encoded ORs apparently down-regulated with heart failure. OR51E1, the most highly expressed of the ORs identified, displayed a more than threefold decrease in failing hearts when compared to control hearts (0.54 × 0.15 vs. 1.75 × 0.48 FPKM). This decrease was independently confirmed by real-time PCR quantification (Figure 2A). However, the other 11 olfactory receptors (OR2A1, OR2A7, OR2A42, OR2C3, OR2H2, OR2I1P, OR5K2, OR7C1, OR10AD1, OR51E2, and OR52N4) were all expressed below 0.1 FPKM and failed to be consistently detected among all samples (Appendix A). Taken together, the results suggest that ectopic OR signaling may be modified by heart failure. However, the low expression of these receptors in cardiac tissue makes it difficult to assess the full extent of the transcriptional remodeling affecting the OR signaling pathway by RNA-Seq.

### 2.3. Real-Time PCR Quantification Reveals Broad Expression of Olfactory Receptors in the Human Heart

We then set out to quantify the expression of 372 genes encoding ORs by real-time PCR. A side-by-side comparison of the results with bulk RNA-Seq data indicated that real-time PCR achieved greater sensitivity for the detection of OR expression in the human heart. In control samples, only ~5% of all ORs screened were detected as expressed at the set minimum level of 0.01 FPKM or higher. With real-time PCR, the ratio of ORs expressed at a meaningful level (Ct < 35) increased to 43% (Figure 2B). Real-time PCR also revealed a trend in the down-regulation of cardiac ORs with ischemic cardiomyopathy. In the failing hearts, the proportion of ORs expressed at low level increased from 24% to 28%, while expression of moderately and highly expressed ORs simultaneously decreased from 19% to 18% (Figure 2B). Six of the ORs that were identified as down-regulated in the failing heart by RNA-Seq were also among the ten most highly expressed cardiac ORs detected by real-time PCR. However, only half of those receptors were confirmed as truly down-regulated (Figure 2C). These results demonstrate that real-time PCR is more adapted for the detection and accurate quantification of mRNAs encoding ectopic ORs in the heart.

### 2.4. Heart Failure Caused by Ischemic Cardiomyopathy Is Characterized by a Specific Ectopic Olfactory Receptor Expression Profile

Principal component analysis of the gene expression data from the 372 ORs quantified by real-time PCR resulted in a clear separation of the four control individuals from the patients suffering from ischemic cardiomyopathy (Figure 3A). A total of 38 ORs were expressed differentially with a 1.4-fold change or greater between control and failing hearts. Expression for approximately two thirds of the receptors (*n* = 26) increased, while the others (*n* = 12) decreased with heart failure (Figure 3B). A heatmap of the relative expression of the 25 most significantly dysregulated ORs for all patients is shown in Figure 3C. Interestingly, 18 out of the 38 differentially regulated receptors were independently found to be expressed in hIPSC-CM and primary cardiomyocytes isolated from human heart ventricles (Figure 3D). This suggests that ectopic OR signaling may be specifically altered in ventricular cardiomyocytes during heart failure.

### 2.5. Altered Cardiac Olfactory Receptor Signaling May Play a Role in the Pathogenesis of Heart Failure

To gain further insight into the biological processes that may be regulated by ectopic OR signaling in the human heart, normalized results from the real-time PCR arrays were integrated with the mapped read counts obtained by RNA-Seq, and the resulting matrix was used for genome-wide co-expression correlation analyses. OR51E1 and OR2B6 were used as examples of ORs expressed in cardiomyocytes that were respectively down-regulated and upregulated with heart failure. Changes in OR51E1 expression were correlated to that of 1419 genes primarily associated with the regulation of mitochondrial biogenesis and respiration (Figure 4A). Querying of the ARCHS4 online repository supported the association of OR51E1 with a cardiac gene set of the tricarboxylic acid cycle and respiratory electron transport (Reactome pathway R-HSA-1428517; *p*adj = 0.006). Changes in OR2B6 expression were correlated to that of 196 other genes predominantly linked to extracellular matrix remodeling (Figure 4B), and querying of the ARCHS4 database further suggested that OR2B6 co-expressed with genes involved in immunoregulatory interactions between immune and non-immune cells (Reactome pathway R-HSA-198933; *p*adj = 0.006). Lastly, we compared the expression pattern of each OR expressed in the heart to that of every other OR to generate the map shown in Figure 4C, as such correlations may help identify GPCRs with related functions. A high-resolution version of the same map is available in Appendix A. Considering the predicted functions of OR51E1 and OR2B6, this map uncovered a cluster of ORs potentially linked to the regulation of mitochondrial function (Figure 4D) and another smaller cluster of receptors possibly involved in the ECM–immune cell crosstalk (Figure 4E). These data suggest that altered cardiac OR signaling may contribute to the development of heart failure.

## 3. Discussion

### 3.1. A Large Number of Ectopic ORs Are Expressed in the Human Heart

We have quantified left ventricular mRNA levels for 372 ORs and provide this dataset as a resource to promote research on the regulation, signaling, and physiological function of these receptors in the heart. Our expression profiling revealed that an unexpectedly high number of OR genes are transcribed in the human heart, with a large proportion of them (>43%) expressed at levels compatible with the presence of biologically functional cell signaling circuits [5,13]. The activation of ectopic ORs has been linked to the induction of G-protein-mediated calcium influx in most cell systems investigated, including in cardiomyocytes and cardiac non-myocyte cells, and can further lead to the modulation of the PI3K/Akt/mTOR, MAPK, and CaMKIV pathways and of the gene expression programs controlled by these kinases [8,9,11]. This suggests that ectopic ORs could be important new players in the regulation of contractile function but also growth and metabolism in the mammalian heart.

### 3.2. Heart Failure Caused by Ischemic Cardiomyopathy Changes the Expression Pattern of Cardiac ORs

Another important finding was that more than three dozen ORs were differentially regulated by heart failure to the extent that diseased patients could be distinguished from normal individuals solely on the basis of their cardiac OR expression profile. This finding may be of pathophysiological relevance considering that mRNA expression of some of the most well-studied and therapeutically targeted non-odorant GPCRs in the heart, including the beta-1 adrenergic receptor and the angiotensin II receptor type 1, is altered with heart failure and correlates with the severity of the disease [14,15]. Several of the differentially regulated ORs were also expressed at similar levels in male and female cardiomyocytes, suggesting at least some degree of overlap in cardiac OR expression between both sexes. It is noteworthy that, while the bulk RNA-Seq analysis detected OR signaling as a biological process potentially dysregulated in the failing human heart, the approach only successfully identified 3 out of the 38 ORs discovered by real-time PCR. Detection of a given transcript with RNA sequencing methods depends on several factors and notably the abundance of the transcript in the sample, the sequencing depth, and the mapping quality. In general, low-expression genes are filtered out prior to comparative transcriptome analyses as a mean to increase differentially expressed gene detection sensitivity [16]. These methodological limitations may explain why ORs are generally absent from large-scale analyses of the transcriptional landscape in heart failure [17].

### 3.3. Ectopic ORs as Regulators of Cardiometabolic Health

Although most ectopic ORs are still orphan, natural ligands are increasingly being discovered amongst the odorant molecules and other metabolites originating from food sources, thus underscoring the importance of those receptors in the maintenance of metabolic homeostasis [9,18]. A unique feature of ORs, which is inherent to the ancestral use of a combinatorial coding scheme to encode odor identities in the olfactory epithelium, is that one odorant molecule can interact with several different ORs with varying affinities and reciprocally one OR can be activated by a range of different ligands [19,20]. For example, 24 compounds with a wide array of chemical functions and structures act as ligands for OR51E1. About 58% of these are carboxylic acids, and the remaining 42% belong to various chemical families encompassing aldehydes, aromatic alcohols, esters, and ethers, as well as nitrogen- and sulfur-containing molecules [21]. Some of these ligands, including short-chain and medium-chain fatty acids, are present at receptor-activating concentrations as circulating factors in plasma and locally in tissues [11]. For example, the binding of nonanoic acid to OR51E1 promotes GLP-1 secretion by the intestinal L-cells and induces a negative chronotropic effect in the cardiomyocyte [11,22], while the mouse ortholog of OR51E1 (Olfr558) located in smooth muscle cells of the renal cortex is strongly activated by butyric acid and isovaleric acid, which may contribute to the regulation of blood pressure [23,24]. Altogether, these observations highlight the complexity of ectopic OR signaling and expose a multitude of potential novel mechanisms by which dietary habits, the microbiome, and metabolic tissues may regulate cardiac function in health and disease.

### 3.4. A Potential Role for Ectopic ORs in Cardiac Remodeling

Gene co-expression correlations provide a robust methodology for predicting gene function, as genes which share a biological process are often co-regulated [25]. The combination of relative gene expression data obtained from bulk RNA-Seq and real-time PCR arrays, together with the pairwise analysis of ectopic OR co-expression, may thus provide novel biological insights for these poorly characterized receptors. Beyond fueling oxidative phosphorylation for energy production, circulating free fatty acids have also recently emerged as critical signaling molecules regulating cardiac antioxidant defenses and inflammation resolution via the free fatty acid receptor 4 (FFAR4), a class A GPCR expressed in cardiomyocytes [26,27]. It is, therefore, highly conceivable that ectopic ORs expressed in the heart participate to the regulation of key biological processes such as mitochondrial respiration, ECM remodeling, or inflammation.

### 3.5. Study Limitations

As general limitation to all transcriptomic analyses, we acknowledge that mRNA expression levels are not always predictive of protein expression levels, nor do they inform on the functionality of the receptor in the tissue considered. While our data suggest that some of the OR genes impacted by heart failure are transcribed in cardiomyocytes, the distribution of these receptors between myocytes and non-myocyte cell populations and their potential alteration with disease states remain to be established. The notorious lack of selectivity of GPCR antibodies will certainly pose a challenge in the further characterization of ectopic ORs expressed in the heart [28].

Another limitation was the relatively small study population, which solely included male patients suffering from heart failure subsequent to ischemic cardiomyopathy. Size of the control group was particularly limited due to the greater difficulty to obtain tissue samples from healthy donors that could yield RNA of sufficient quality for the analyses. While unsupervised analyses identified clear changes in cardiac gene expression signature between control and diseased samples, future investigations should aim to determine whether cardiac OR expression is affected by sex, age, race, or other etiological causes or stages of heart failure. Given the association of several GPCR polymorphisms with the development of heart failure, the pathophysiological role of allelic mutations in genes encoding functionally relevant cardiac ORs would also be worth investigating [29]. Lastly, and although conserved orthologous ectopically expressed ORs are evolutionary constrained, the lack of correlation in the tissue expression profiles between some of the human and mouse OR orthologous pairs may further complicate the identification of their physiological functions in animal systems [30,31,32]. A direct comparison of our dataset with results from a murine ectopic OR gene expression profiling should facilitate rational use of the mouse to model the roles of ORs in cardiac physiology and disease.

## 4. Materials and Methods

### 4.1. Human Heart Samples

We studied deidentified left ventricular cardiac tissue samples from 26 patients diagnosed with end-stage heart failure with reduced ejection fraction who were referred to the Texas Heart Institute for heart transplantation and placed on Left Ventricular Assist Device (LVAD) support. All selected patients were diagnosed with ischemic cardiomyopathy. Left ventricular tissue was obtained from the apex during LVAD implantation, flash frozen in liquid nitrogen, and stored in an ultra-low-temperature freezer until total RNA extraction. Deidentified, normal left ventricular heart tissue samples collected post mortem from 7 male individuals were purchased from the Reprocell Bioserve Global Biorepository (Beltsville, MD, USA). A total of 11 samples (8 failing heart and 3 normal control samples) were later excluded from the study due to insufficient RNA quantity (*n* = 1) or poor RNA quality (RNA Integrity Number (RIN) < 5; *n* = 10). The average age at time of sample procurement was 47.2 years for the remaining control individuals (*n* = 4) and 60.5 years for heart failure patients (*n* = 18). Age and additional demographic and clinical characteristics of the selected patients are provided in Appendix A.

### 4.2. Human Cardiomyocyte Models

Primary human ventricular cardiac myocytes preserved in RNAlater (Cat. No. C-14080) were purchased from PromoCell GmbH (Heidelberg, Germany). Four vials of 1 million cells each from the same lot (No. 475Z017.2P-3) were pooled prior to proceeding with RNA extraction as described below. Cells were harvested from a 22-year-old Caucasian male donor. Human cardiomyocytes derived from induced pluripotent stem cells (hiPSC-CMs) were purchased from FUJIFILM Cellular Dynamics, Inc. (Madison, WI, USA). Cells from two backgrounds with no known disease-related genotypes were used in parallel for the experiments. Cells from female donors were used to complement analyses of male cells and heart samples. Both donors No. 01434 and No. 11713 were healthy Caucasian females of less than 39 years of age at time of sampling. The hiPSC-CMs were plated according to the manufacturer’s instructions and cultured in maintenance medium for 7 days prior to RNA extraction.

### 4.3. RNA Extraction and RNA Quality Control

A total of 80 to 100 mg of human ventricular heart tissue was homogenized in TRIzol reagent with an Omni Bead Ruptor bead mill homogenizer (Omni International, Kennesaw, GA, USA). Total RNA was purified from the lysed samples using PureLink spin column RNA isolation kit (Invitrogen, Waltham, MA, USA). Genomic DNA removal was performed in two steps, first by on-column DNA digestion with PureLink DNase during the RNA purification step and, next, by proceeding with a rigorous DNase treatment using TURBO DNA-free kit (Invitrogen) following RNA elution. Total RNA concentration was measured by NanoDrop and RIN determined with an Agilent 2100 Bioanalyzer Instrument (Agilent Technologies, Santa Clara, CA, USA). Total RNA from cells was extracted with TRIzol reagent and subjected to DNase treatment with turbo DNA-free as conducted for the human heart tissue samples.

### 4.4. Bulk RNAsequencing

Complete RNA-Seq workflow, from RNA library preparation to estimation of gene expression level, was carried out by Singulomics Corporation (Bronx, NY, USA). In brief, the RNA libraries were developed using NEBNext Ultra II RNA kit (New England Biolabs, Ipswich, MA, USA). The library of pooled samples was sequenced using Illumina NovaSeq 6000 system on a PE150, 20 million reads, sequencing run (Illumina, San Diego, CA, USA). Following filtering of raw reads, mapping of clean reads to the reference genome was accomplished using HISAT2 software version 2.2.1 [33]. Fragments per kilobase of transcript per million mapped reads (FPKM; Appendix A) were used to estimate gene expression levels.

### 4.5. Real-Time PCR Quantification of Olfactory Receptors

Custom 384-well PCR plates were designed with PrimePCR SYBR Green assays (Bio-Rad, Hercules, CA, USA) to quantify mRNA levels of 372 distinct olfactory receptors. For each sample, 11 µg of total RNA was reverse transcribed with SuperScript IV reverse transcriptase (Invitrogen) by performing 11 RT reactions of 1 µg RNA each in parallel. Next, cDNAs were pooled and transferred to the wells of the custom arrays (25 µg/well) and amplified with SsoAdvanced Universal SYBR Green Supermix on a CFX384 Touch real-time PCR detection system (Bio-Rad). All arrays passed quality control criteria for qPCR performance, RNA quality, and genomic DNA contamination of samples. Gene expression quantification was performed with the delta Ct (2^−ΔCt^) method, using the geometric mean of housekeeping genes encoding Peptidylprolyl Isomerase A (Cyclophilin A; PPIA) and Tyrosine 3-Monooxygenase/Tryptophan 5-Monooxygenase Activation Protein Zeta (YWHAZ) for normalization. Information about the array setup, raw cycle threshold (Ct) values for each sample, and calculations can be found in Appendix A.

### 4.6. Data Analysis

Comprehensive analysis of normalized gene expression data was carried out with freely available web-based interfaces [34,35]. In brief, principal component analysis (PCA) and correlation heatmaps with hierarchical clustering were used to visualize similarities and dissimilarities between individual samples and to organize and analyze gene–gene interactions on a global scale. Gene Set Enrichment Analysis based on Reactome pathway database was implemented on bulk RNA-Seq data to identify biological processes that were differentially regulated between failing and normal hearts. The ORs differentially regulated between failing and normal hearts were identified by univariate analysis (volcano plot) and organized/visualized for each individual sample by hierarchical clustering heatmap. In order to predict biological functions for specific ORs, normalized expression data from the real-time PCR arrays were integrated with the RNA-Seq FPKM values, and the resulting expression matrix (Appendix A) was used to determine genes with significantly correlated expression. The web interface Correlation AnalyzeR was used to further explore co-expression correlations and predict gene functions using cardiac datasets from the ARCHS4 repository [25]. All other graphical representations and statistical analyses were performed with GraphPad Prism 9. Unless otherwise indicated, *p* < 0.05 was considered statistically significant.

## Figures and Tables

**Figure 1 ijms-24-13709-f001:**
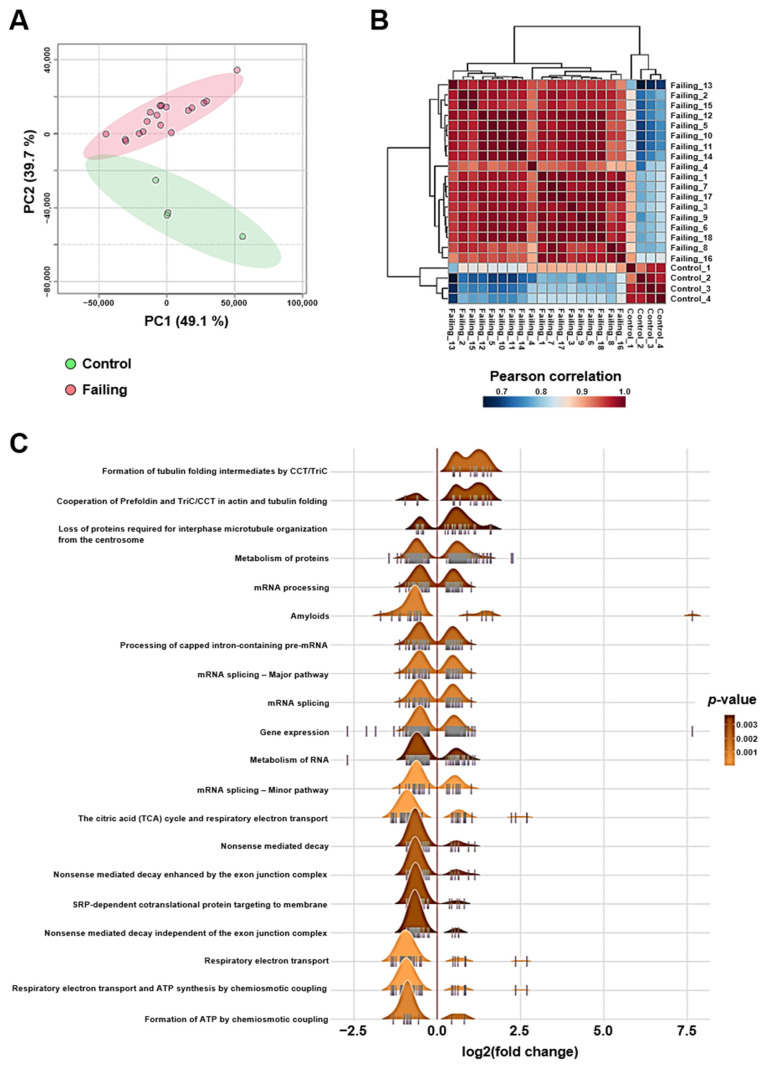
Failing human hearts are characterized by altered proteostasis, mRNA splicing, and bioenergetics. (**A**) Unsupervised classification using PCA on the cardiac global gene expression profiles generated by bulk RNA sequencing. (**B**) Correlation matrix between the samples. The heatmap matrix shows the Spearman correlation coefficient between samples for all expressed genes measured by bulk RNA sequencing. Samples cluster by phenotype. Red colors represent relationships between samples that are most similar; blue colors represent samples that are more dissimilar with lower coefficients. (**C**) Ridgeline plot visualizing expression distribution of core enriched genes for the top 20 GSEA enriched categories.

**Figure 2 ijms-24-13709-f002:**
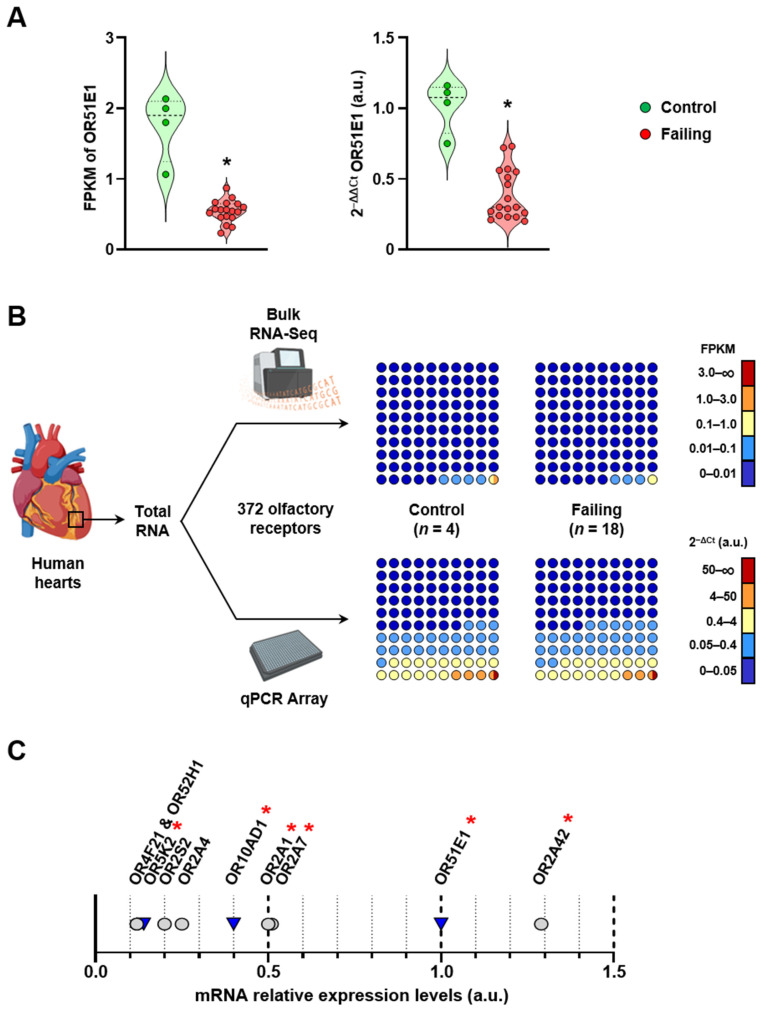
Real-time PCR provides a more sensitive and accurate overview of OR expression in the human heart. (**A**) Comparison of mRNA expression levels for OR51E1 as determined by RNA sequencing (FPKM) and real-time PCR quantification using the delta-delta Ct method. * *p* < 0.05. (**B**) Comparison of the expression levels of 372 ORs as determined by RNA sequencing (FPKM) and real-time PCR quantification using the delta Ct method. Blue colors represent low expression; yellow indicates moderate expression; orange and red colors indicate high expression. Each dot of the 10 × 10 dot plot represents 1% of all ORs quantified. A 2^−ΔCt^ value of 0.05 or higher corresponds to a Ct < 35. (**C**) Relative expression levels of the 10 most highly expressed ORs in the non-failing human left ventricle as determined by real-time PCR quantification. Grey circles represent ORs with expression unchanged in the failing heart. Blue triangles represent ORs that are downregulated in the failing heart. Red asterisks indicate ORs detected as downregulated in the failing heart according to bulk RNA-Seq analysis.

**Figure 3 ijms-24-13709-f003:**
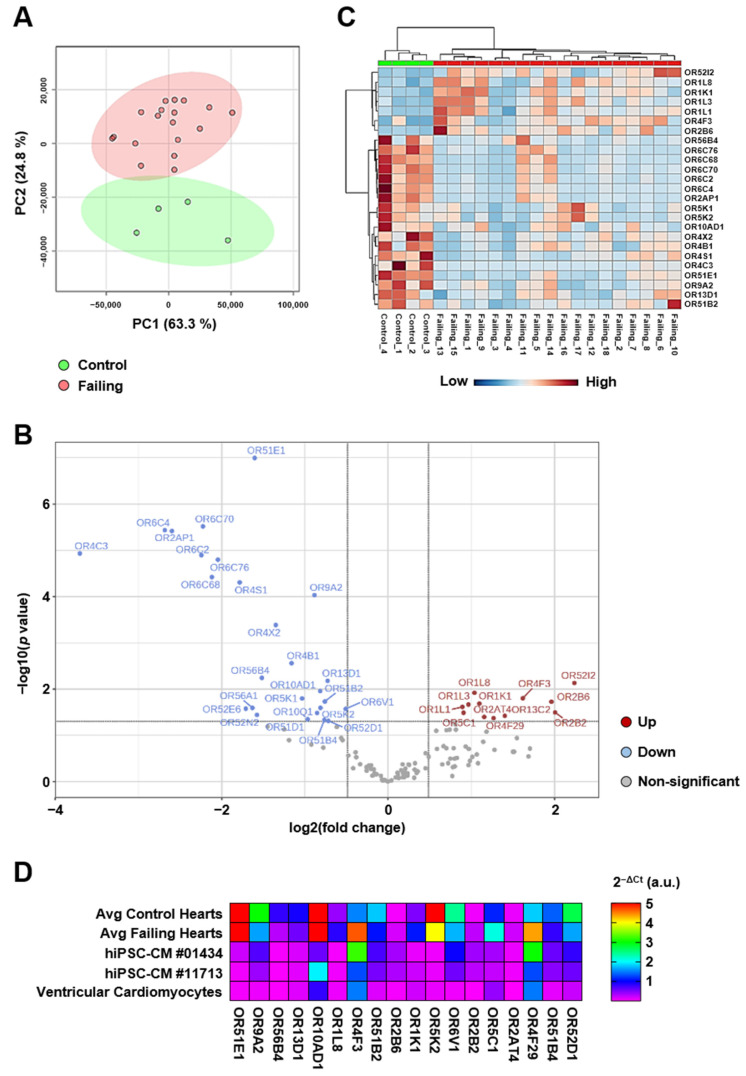
Heart failure changes the ectopic OR gene expression signature. (**A**) Unsupervised classification using PCA on the cardiac OR gene expression profiles generated by real-time PCR array. (**B**) Volcano plot visualization of the 38 ORs that are either increased (red) or decreased (blue) by 1.4-fold or higher in the failing hearts when compared to non-failing control hearts. (**C**) Hierarchical-cluster heatmap representing the expression levels of the top 25 ORs that are differentially regulated between failing and non-failing hearts. Blue colors represent low expression; red colors indicate high expression. (**D**) Heatmap visualization of the expression levels of the 18 ORs differentially regulated between failing and non-failing hearts that were also detected in human cardiomyocytes derived from induced pluripotent stem cells (hiPSC-CMs) and primary human ventricular cardiac myocytes.

**Figure 4 ijms-24-13709-f004:**
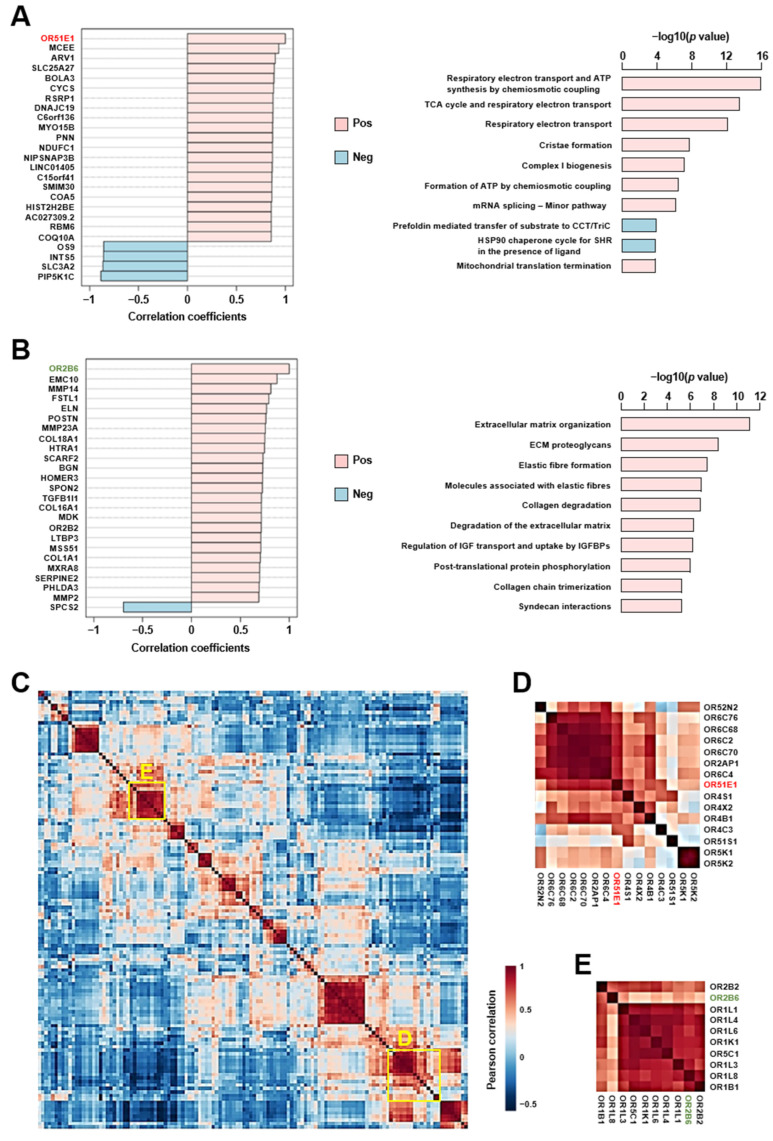
Olfactory receptors may play a role in cardiac remodeling during heart failure. (**A**) Representation of the top 24 genes out of a total of 1419 differentially regulated genes (DRG) that are correlated to OR51E1 downregulation in the failing human heart (*p* < 0.01). OR51E1 is shown in red. The top 10 gene ontology (GO) biological processes linked to those 1419 DRG are also presented. Pink bars represent genes and GO biological processes that are also down-regulated in the failing heart. Blue bars represent genes and GO biological processes that are upregulated in the failing heart. (**B**) Representation of the top 24 genes out of a total of 196 DRG that are correlated to OR2B6 upregulation in the failing human heart (*p* < 0.01). OR2B6 is shown in green. The top 10 GO biological processes linked to those 196 DRG are also presented. Pink bars represent genes and GO biological processes that are also upregulated in the failing heart. Blue bars represent genes that are down-regulated in the failing heart. (**C**) Hierarchical-cluster correlation heatmap generated on the basis of Pearson r coefficients calculated for interactions between each OR with all others. A thumbnail image is shown here, and a full-size image is available in Appendix A. Receptors with similar expression patterns are shown in red; distinct expression patterns are shown in blue. The x and y axes are mirror images of one another. The diagonal represents each OR interacting with itself (perfect similarity in expression). Clustering of receptors by similarity of expression reveals a group of ORs associated with OR51E1 (in red) that may regulate mitochondrial function (**D**) and ORs clustering with OR2B6 (in green) that may be associated with extracellular matrix remodeling and the regulation of immune cell function (**E**).

## Data Availability

Source data for this study are openly available on figshare: https://doi.org/10.6084/m9.figshare.23664216 (accessed on 5 July 2023).

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
