# Peer review of "A Two-Step Transcriptome Analysis of the Human Heart Reveals Broad and Disease-Responsive Expression of Ectopic Olfactory Receptors"

_ijms, 2023, doi:10.3390/ijms241813709_

Round 1

Reviewer 1 Report

This manuscript aims to identify the gene expression pattern of Ectopic Olfactory Receptors (ORs) in left ventricular heart tissue. It is interesting, and of course quite important, to understand how the gene is changed after heart failure. Such understanding would be helpful in developing new therapeutic strategy in treating the heart disease. This manuscript, in general, would be an interesting paper for the community.

I am not an expert on the transcriptomics methodology, so the -omics methods used in this manuscript has to be assessed by other reviewers.

I have only one major concern which is about the sample number for the non-failing heart tissue. Comparing to the n number (18) of heart failure samples, the non-failing heart samples have a small n number (4). Even these 4 samples have a big data variation (e.g. in Figure 2), which makes one wonder how accurate, how meaningful and how representative these data are! As this set of data works as an important control, it is important to address this issue and make sure that the identified gene expression change really takes place in the heart failure tissue rather than just the methodology issue arising from sample size. 

Author Response

We thank the reviewer for the encouraging words. We hope that our replies to your comments will adequately address all concerns.

We acknowledge the relatively small size of our control group. Unfortunately, it is much more difficult to obtain ventricular samples from seemingly healthy heart donors than it is to collect them from heart failure patients. This often leads to such comparative studies including a much larger group of heart failure samples.

In order to confirm the phenotypic classification of our study samples, we performed multiple unsupervised analyses (including PCA in figures 1A and 3A and hierarchical clustering in figures 1B and 3C). In each case, there was a clear separation between the 4 controls and the 18 heart failure samples, which independently confirmed grouping of the samples based on their phenotypic variance.

We nonetheless agree that the small size of our study population doesn’t allow for a detailed comparison of gene expression profiles based on parameters such as sex, age, race, or various etiological causes of heart failure, and we have added this limitation in the Discussion, page 11, lines 281-286:“Another limitation was the relatively small study population which solely included male patients suffering from ischemic cardiomyopathy, and 4 Caucasian individuals for the control group. While unsupervised analyses identified clear changes in cardiac gene expression signature between control and diseased samples, future investigations should aim to determine whether cardiac OR expression is affected by sex, age, race, or other etiological causes of heart failure.”

Reviewer 2 Report

The authors of the current manuscript present results from an original study, by which they investigate the role of ectopic olfactory receptors (ORs) in patients with heart failure (HF). This topic is very important from scientific and clinical point of view since about 1-2% of the general population is affected by this syndrome (>10% of people aged 70 and over). Despite the modern pharmacological and non-pharmacological options, the prognosis of HF remains poor (a progressive course of the disease) with >40% of HF patients dying withing 5 years after the diagnosis has been established. So, elaboration of new treatment strategies targeting additional mechanisms of development/progression of HF will be valuable. The  manuscript is very interesting and I believe it will attract reader's attention. 

I have the following recommendations to the authors:

1. Please, put "Materials and methods" after "Introduction" and before "Results"

2. It is worth specifying the characteristics of HF patients, whose hearts you have examined, if this information is available/accessible. Results could be associated and/or varying with patient's characteristics. What is known from the current version of the manuscript is that patients have ischemic heart disease (IHD). You should also point out the age, gender, concomitant risk factors for HF (beside IHD), eventually duration of HF, treatment. 

3. It is mandatory to specify if HF is with preserved (HFpEF), mildly reduced or reduced left-ventricular ejection fraction (HFpEF). The leading etiologies and pathogenesis of HFpEF and HFrEF differ and it could exert influence on the investigated parameters.

4. Could we be sure that the findings about the investigated parameters are due only to HF, but not also to the preceding (sometimes for years) ischemic heart disease? Could patients with IHD without HF have similar findings..? It is worth comparing at a next stage of your study different subpopulations of HF patients.

5. If etiology of HF in all patients from the study population is IHD, it should be specified in the article title. Title should be generally changed, so that it becomes clear the article is about ORs characteristics in HF patients (or patients with ischemic cardiomyopathy).

6. I recommend to the authors to add "Study limitations" after "Discussion".

Author Response

Authors’ response: We thank the reviewer for highlighting the specific research and clinical interests of our study.

  1. Please, put "Materials and methods" after "Introduction" and before "Results"

Authors’ response: The manuscript has been formatted accordingly to the journal’s style. In articles published in International Journal of Molecular Sciences, the Materials and Methods section comes after the Discussion.

  1. It is worth specifying the characteristics of HF patients, whose hearts you have examined, if this information is available/accessible. Results could be associated and/or varying with patient's characteristics. What is known from the current version of the manuscript is that patients have ischemic heart disease (IHD). You should also point out the age, gender, concomitant risk factors for HF (beside IHD), eventually duration of HF, treatment.

Authors’ response: Patients demographics and other relevant information that was accessible to us are provided as a supplemental file in Dataset S2. The data include age, gender, ethnicity, date of tissue collection (i.e. date of LVAD implantation for heart failure patients, and date of autopsy for controls), BMI, and RNA Integrity Number (RIN). We also provide the post mortem interval (PMI), cause of death, and comorbidities for control individuals. We didn’t have access to comorbidities, HF duration, or treatment information for the heart failure patients.

Dataset S2 is referenced in section 4.1 of the manuscript (page 11) and provided in the supplemental files accompanying the manuscript. Dataset S2 is also accessible through figshare following the link provided in the Data Availability Statement (page 13).

  1. It is mandatory to specify if HF is with preserved (HFpEF), mildly reduced or reduced left-ventricular ejection fraction (HFpEF). The leading etiologies and pathogenesis of HFpEF and HFrEF differ and it could exert influence on the investigated parameters.

Authors’ response: We have modified section 4.1 of the manuscript to provide this information (page 10): “We studied de-identified left ventricular cardiac tissue samples from 26 patients diagnosed with end-stage heart failure with reduced ejection fraction who were referred to the Texas Heart Institute for heart transplantation and placed on Left Ventricular Assist Device (LVAD) support. All selected patients were diagnosed with ischemic cardiomyopathy.”

  1. Could we be sure that the findings about the investigated parameters are due only to HF, but not also to the preceding (sometimes for years) ischemic heart disease? Could patients with IHD without HF have similar findings? It is worth comparing at a next stage of your study different subpopulations of HF patients.

Authors’ response: This is a very good point. It is definitely possible that the expression of at least some of the cardiac ORs changes over the time course of heart failure progression. We have modified the section discussing the study limitations to add this point as a next step in our studies (page 10): “While unsupervised analyses identified clear changes in cardiac gene expression signature between control and diseased samples, future investigations should aim to determine whether cardiac OR expression is affected by sex, age, race, or other etiological causes or stages of heart failure.”

  1. If etiology of HF in all patients from the study population is IHD, it should be specified in the article title. Title should be generally changed, so that it becomes clear the article is about ORs characteristics in HF patients (or patients with ischemic cardiomyopathy).

Authors’ response: We prefer to leave the title as is in order to keep mention of profiling OR gene expression in the normal human heart, which was our first goal prior to identifying ORs dysregulated by heart failure from ischemic origin. However, we modified the text in several places to make it clearer that we focused on heart failure caused by ischemic cardiomyopathy:

  • Abstract, page 1: “The goals of this study were to profile OR gene expression in the human heart, to identify ORs dysregulated by heart failure caused by ischemic cardiomyopathy, (…)”
  • Introduction, page 2: “The goals of the present study were to generate a high-resolution map of OR gene expression in the human left ventricle, to identify ORs whose expression is altered in response to heart failure caused by ischemic cardiomyopathy, and (…)”
  • Results, page 5: “2.4 Heart Failure Caused by Ischemic Cardiomyopathy is Characterized by a Specific Ectopic Olfactory Receptor Expression Profile”
  • Discussion, page 9: “3.2 Heart Failure Caused by Ischemic Cardiomyopathy Changes the Expression Pattern of Cardiac ORs”

  1. I recommend to the authors to add "Study limitations" after "Discussion".

Authors’ response: We have added sub-sections to the Discussion, with the last sub-section labelled as “3.5 Study Limitations”.

Reviewer 3 Report

Interesting study using transcriptomics to investigate the presence of olfactory receptors in cardiac tissue. Few cases but with relevant data. In the discussion they can add the relevance of mutated alleles in cases of cardiomyopathy and its implications in the evolution of heart failure.

Limitations should include whether the patients chosen were of the same ethnicity and with similar cardiac dysfunction mechanism.

Author Response

We thank the reviewer for the constructive comment. All heart failure patients included in this study were diagnosed with ischemic cardiomyopathy. There were 9 Caucasian, 2 African American, and 7 Hispanic patients. The 4 controls were all Caucasian individuals.

This information on the cardiac dysfunction mechanism and patients’ ethnicity is provided in the supplementary Dataset S2, as mentioned in the Materials and Methods section, page 12, lines 309-310: “Age and additional demographic and clinical characteristic of the selected patients are provided in Dataset S2”.

We have also expanded the limitations section of the Discussion on page 11, lines 281-289, as follows: “Another limitation was the relatively small study population which solely included male patients suffering from ischemic cardiomyopathy, and 4 Caucasian individuals for the control group. While unsupervised analyses identified clear changes in cardiac gene expression signature between control and diseased samples, future investigations should aim to determine whether cardiac OR expression is affected by sex, age, race, or other etiological causes of heart failure. Given the association of several GPCR polymorphisms with the development of heart failure, the pathophysiological role of allelic mutations in genes encoding functionally relevant cardiac ORs would also be worth investigating [29].”

Round 2

Reviewer 1 Report

The authors have addressed my concerns.

Please include the information about why the sample size for healthy group is so small in the main texts for the interests of the readers. 

Author Response

We have modified the section discussing the study limitations to include this information. The text now reads as follows (page 10): “Another limitation was the relatively small study population which solely included male patients suffering from heart failure subsequent to ischemic cardiomyopathy. Size of the control group was particularly limited due to the greater difficulty to obtain tissue samples from healthy donors that could yield RNA of sufficient quality for the analyses.”